# Infrared and 2-Dimensional Correlation Spectroscopy Study of the Effect of CH$_3$NH$_3$PbI$_3$ and CH$_3$NH$_3$SnI$_3$ Photovoltaic Perovskites on Eukaryotic Cells

Luca Quaroni [1,*], Iness Benmessaoud [2], Bertrand Vileno [3], Endre Horváth [2] and László Forró [2,*]

1   Department of Physical Chemistry and Electrochemistry, Faculty of Chemistry, Jagiellonian University, PL-30387 Kraków, Poland
2   Laboratory of Physics of Complex Matter, Ecole Polytechnique Fédérale de Lausanne (EPFL), 1015 Lausanne, Switzerland; ines.benmess@gmail.com (I.B.); endre.horvath@epfl.ch (E.H.)
3   POMAM Lab, Institut de Chimie de Strasbourg UMR 7177, CNRS/UdS. 4 rue Blaise Pascal, CS 90032, 67081 Strasbourg CEDEX, France; vileno@unistra.fr
*   Correspondence: luca.quaroni@uj.edu.pl (L.Q.); laszlo.forro@epfl.ch (L.F.)

**Abstract:** We studied the effect of the exposure of human A549 and SH-SY5Y cell lines to aqueous solutions of organic/inorganic halide perovskites CH$_3$NH$_3$PbI$_3$ (MAPbI$_3$) and CH$_3$NH$_3$SnI$_3$ (MASnI$_3$) at the molecular level by using Fourier transform infrared microspectroscopy. We monitored the infrared spectra of some cells over a few days following exposure to the metals and observed the spectroscopic changes dominated by the appearance of a strong band at 1627 cm$^{-1}$. We used Infrared (IR) mapping to show that this change was associated with the cell itself or the cellular membrane. It is unclear whether the appearance of the 1627 cm$^{-1}$ band and heavy metal exposure are related by a direct causal relationship. The spectroscopic response of exposure to MAPbI$_3$ and MASnI$_3$ was similar, indicating that it may arise from a general cellular response to stressful environmental conditions. We used 2D correlation spectroscopy (2DCOS) analysis to interpret spectroscopic changes. In a novel application of the method, we demonstrated the viability of 2DCOS for band assignment in spatially resolved spectra. We assigned the 1627 cm$^{-1}$ band to the accumulation of an abundant amide or amine containing compound, while ruling out other hypotheses. We propose a few tentative assignments to specific biomolecules or classes of biomolecules, although additional biochemical characterization will be necessary to confirm such assignments.

**Keywords:** infrared spectroscopy; infrared microscopy; 2DCOS; perovskite; amyloid

## 1. Introduction

The perovskite methylammonium lead iodide CH$_3$NH$_3$PbI$_3$ (MAPbI$_3$) has been the subject of interest for photovoltaic (PV) applications due to its elevated light conversion efficiency [1]. However, its use is associated with the potential toxicity arising from lead [2–5], particularly because of the potential human exposure during the production process. This concern has driven the search for a replacement for Pb with reduced toxicity and lower environmental impact, without compromising the efficiency of MAPbI$_3$. The structurally similar perovskite CH$_3$NH$_3$SnI$_3$ (MASnI$_3$), containing tin instead of lead, has been proposed as a viable candidate. Despite its appeal, based on its efficiency in light conversion, the compound also bears potential hazards due to the toxicity of tin [6]. For this reason, we decided to perform a systematic comparison of the effects of cellular exposure to Pb and Sn containing perovskites. Previous work by some of the authors has described the in vitro cytotoxic effects on eukaryotic cells exposed to MAPbI$_3$ [2]. In this follow-up work, we use infrared

(IR) absorption spectroscopy as a method for the comparison of the molecular changes that accompany exposure to both MAPbI$_3$ and MASnI$_3$.

IR spectroscopy has been used for decades to characterize the molecular properties of biological samples. Molecular species absorb infrared light, giving rise to absorption patterns that reflect the composition of the sample. Spectral recordings can be obtained from samples of any composition without the need of chemical modification such as molecular labeling or staining. Analysis of complex biological samples can be performed using standard preparations such as fixed cells and tissue sections [7]. With some challenges, the measurements can be extended to living cells [7]. In some cases, the compositional information from IR spectra can be complemented by molecular information. IR absorption patterns are sensitive to structural parameters such as molecular conformation, bond length, and bond strength. One application of this capability is the study of protein secondary structure and, in some cases, of tertiary structure [8,9]. In more recent years, applications of IR spectroscopy were boosted by the coupling of interferometric technology for spectral analysis and optical microscopy configurations, leading to the introduction of Fourier transform infrared microspectroscopy (FTIRMS). FTIRMS allows for the IR spectra of single cells to be recorded with sufficient sensitivity to provide detailed analysis of molecular properties [10,11]. The applications extend beyond fundamental research in cell biology and include biomedical research and molecular diagnostics [12]. The coupling of IR microscopes to synchrotron light sources has provided improved performance, enabling high signal-to-noise measurements with diffraction-limited spatial resolution [13]. These developments have extended FTIRMS to the study of weaker spectroscopic features in cellular samples. Despite these advances, one remaining challenge is the assignment of absorption bands in complex cellular spectra to specific molecular absorbers. Assignments based on functional group identification have often been used in the recent literature. However, it has been stressed that such an approach is often unreliable for the purpose of compound identification, given the extensive number of cellular absorbers that provide overlapping bands in the same spectral region. It was proposed by one of these authors that the complexity of the problem could be reduced by using 2D correlation spectroscopy (2DCOS) to analyze the correlations between multiple bands in cellular spectra that evolve over time and thus identify bands arising from the same absorber. In the present work, we demonstrate for the first time the application of 2DCOS to the assignment of bands in static samples by using the spatial position of the measurement as the physical variable along which the spectra are aligned. We use this approach to interpret the molecular origin of the spectroscopic changes observed in the sample following metal exposure.

## 2. Results and Discussion

We studied the response of the SH-SY5Y neuroblastoma cell line and the A549 pulmonary carcinoma cell line following exposure to MAPbI$_3$ and MASnI$_3$. Both MAPbI$_3$ and MASnI$_3$ are water-soluble compounds, and their dissolution in the culture medium provides a variety of aqueous forms of Pb$^{2+}$ and Sn$^{2+}$ ions, in addition to insoluble species such as lead carbonates, hydroxides, and phosphates. Experimental details of the preparation are provided in Section 4. In this study, we removed the precipitates before incubating the cells with the conditioned medium. Therefore, we ascribe the biological effects to soluble species such as aqueous Pb$^{2+}$ and Sn$^{2+}$, and possibly to methylammonium cations and iodine derivatives. Mortality and overall response in cell lines exposed to either MAPbI$_3$ or MASnI$_3$ suspensions showed an increase in cellular mortality, thus underlining the toxicity of both compounds [6]. An example of the toxicity response of the two cell lines is provided in Appendix A (Figure A1). Despite the comparable increase in cell mortality, toxicity of MAPbI$_3$ seems to be mediated by different mechanisms in the SH-SY5Y and A549 cell lines [2].

We tried to gain insight into the molecular events that accompany cellular response to metal exposure by performing FTIRMS measurements on fixed adherent cells following treatment with the perovskites. The cells were cultured on CaF$_2$ optical slides and inspected using an FTIR microscope to collect transmission IR absorption spectra and maps of single cells with diffraction-limited spatial

resolution. The FTIR absorption spectrum from a SH-SY5Y cell in the energy range 1000–4000 cm$^{-1}$ is shown in Figure 1A. Figure 1B shows the details of the spectrum of a single cell exposed to 100 µg/mL of MAPbI$_3$ (blue) or 100 µg/mL of MASnI$_3$ (red) and compares it to the spectrum of a non-treated cell (black).

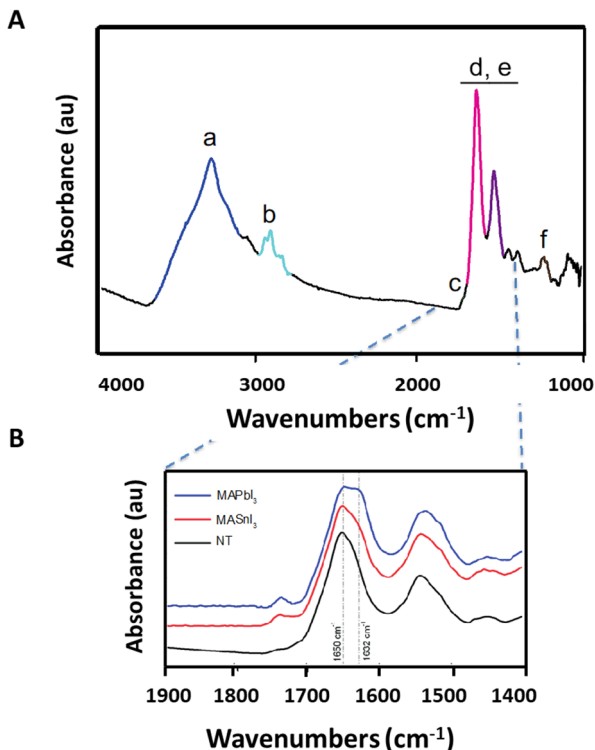

**Figure 1.** Infrared (IR) spectra of the SH-SY5Y neuroblastoma cells. (**A**) IR spectrum from a SH-SY5Y cell. (**B**) Zoom-in on the absorbance in the 1350–1750 cm$^{-1}$ range of non-treated (black), MAPbI$_3$ (blue), and MASnI$_3$ (red) treated cells. Figure reproduced and modified from [6] (I.B. doctoral thesis).

The main contribution to the spectral region detailed in Figure 1B arises from the absorption bands of carbonyl and imide containing compounds, although contributions from other organic molecules such as amines are also present. In cellular spectra, the dominant contribution in this region is usually assigned to polypeptide bands, termed Amide I and Amide II, around 1650 cm$^{-1}$ and 1550 cm$^{-1}$, respectively, with the assumption that protein absorption is dominant. In purified proteins, the fine structure of these bands is often used to quantify the relative abundance of specific secondary structure components [14]. The use of amide bands for the study of protein conformation has also been extended to cells and tissue. Application to such samples is challenging and limited by the compositional complexity of the samples themselves. Cells and tissues, particularly live ones, are characterized by a mixture of polypeptides and other organic molecules absorbing in the same spectral region. In many cases, performing a detailed analysis of protein secondary structure is unreliable in such samples because of overlapping contributions from multiple absorbing molecules [15]. Despite these difficulties, some cases have been reported where the Amide I band has been used to track protein conformational changes within cells or tissues. One such case is the study of neurodegenerative diseases involving the formation of amyloid deposits, consisting of misfolded polypeptides that aggregate into an intermolecular β-sheet conformation. Since the deposits are compositionally nearly pure, they have been identified from the characteristic Amide I absorption at ca. 1625–1628 cm$^{-1}$ [10,16,17]. In early examples, these assignments were confirmed using staining protocols for amyloids. In the case of fixed cells and tissue, the contribution of many small molecules to this spectral region is removed by the washing steps of fixation, thus simplifying the problem of interference from small molecule absorption.

To facilitate the resolution of different overlapping spectral contributions and reduce baseline effects, we analyzed the second derivative of the absorbance spectra. The 2nd derivative traces in the $1200 \text{ cm}^{-1}$–$1800 \text{ cm}^{-1}$ spectral region for cells unexposed and exposed to perovskite suspensions are shown in Figure 2. The traces show the 2nd derivative of the average spectra collected by mapping single cells at a diffraction-limited spatial resolution. Figure 2A shows the effect of exposure to MAPbI$_3$ and MASnI$_3$ on the spectra of A549 cells. Figure 2B shows the corresponding measurements performed on metal treated and untreated SH-SYS5 cells. Spectra in both A and B were recorded on cells fixed after five days of exposure to the metals. Figure 2C shows the difference between the average spectrum of A549 cells exposed to MAPbI$_3$ and unexposed cells to highlight dominant spectral changes.

For unexposed cells, the Amide I region of unexposed cells is characterized by a strong contribution at $1655 \text{ cm}^{-1}$ and a shoulder at $1638 \text{ cm}^{-1}$. This pattern is common to many eukaryotic cell types and is due to the abundance of $\alpha$-helical proteins absorbing around $1655 \text{ cm}^{-1}$. The contribution from $\beta$-sheet protein structures, at approximately $1635 \text{ cm}^{-1}$, is responsible for the shoulder on the lower wavenumber side. Other secondary structure components absorb in this region, together with contributions from biomolecules other than proteins, and induce small variations on this basic pattern. A weaker contribution is also observed at $1680$–$1690 \text{ cm}^{-1}$, which can arise from other secondary structure motifs such as $\beta$-sheet, $\beta$-turns, and other molecules including nucleic acids and sphingolipids. Due to the preparation protocol of the samples involving repeated washings following fixation, it is likely that contributions from small water soluble and alcohol soluble molecules such as metabolites have been removed, except for acyl lipids, which are at least partially retained by fixation.

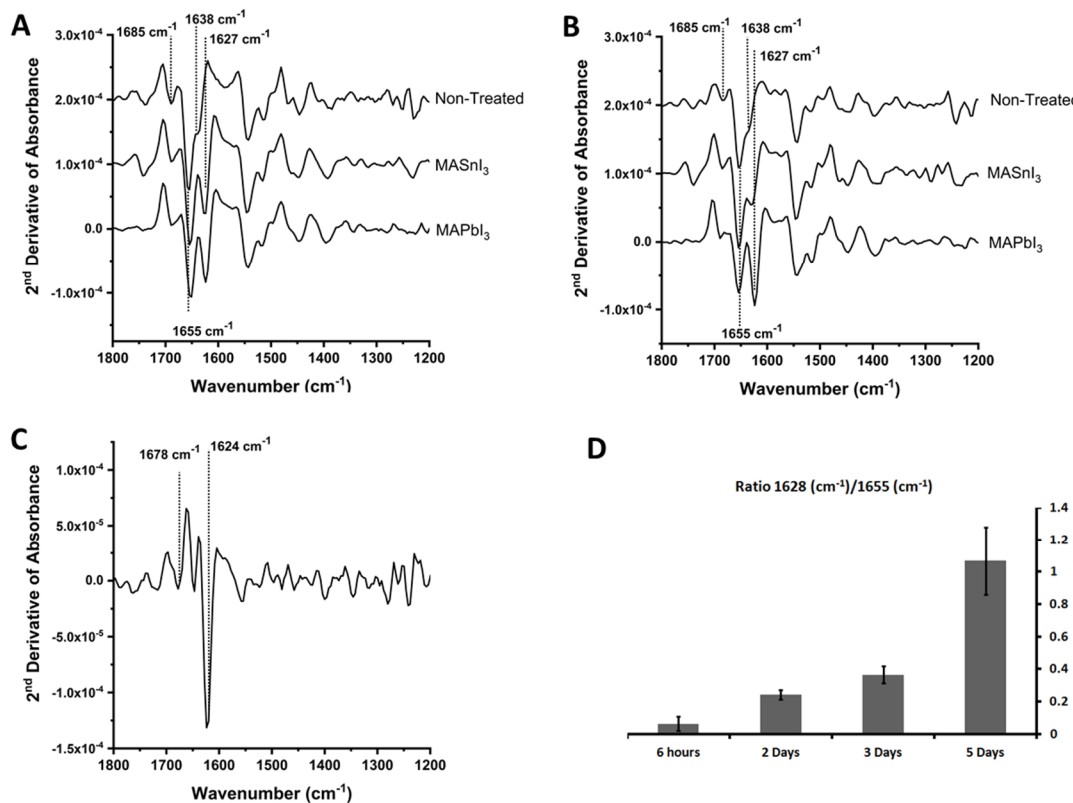

**Figure 2.** Average 2nd derivative of IR absorbance spectra of single cells in the $1200 \text{ cm}^{-1}$–$1800 \text{ cm}^{-1}$ range. (**A**) A549 cells, unexposed and after five days exposure to MAPbI$_3$, MASnI$_3$. (**B**) SH-SYS5 cells, unexposed and after five days exposure to MAPbI$_3$ and MASnI$_3$. (**C**) Subtraction between the average spectrum of A549 cells, exposed and unexposed to MAPbI$_3$. (**D**) Time course of the ratio of 2nd derivative absorption at $1652 \text{ cm}^{-1}$ and $1627 \text{ cm}^{-1}$ following exposure of A549 cells to MAPbI$_3$, expressed as average $\pm \sigma$. Figure reproduced and modified from [6] (I.B. doctoral thesis).



Figure 2 shows that exposure to both metal salts causes the appearance of a new strong band in the Amide I region with an apparent maximum at 1627 cm$^{-1}$. The subtraction spectrum of Figure 2C highlights the spectral changes and confirms that the main change is the appearance of a band with a maximum at 1624 cm$^{-1}$, accompanied by other minor components. The difference between the apparent band maxima in the total spectra and subtraction spectra may be due to the presence of overlapping bands that shift the apparent band maximum. Changes are also seen in the Amide II region, with some components disappearing and new components appearing after exposure to metals. The Amide II band is also sensitive to the secondary structure of polypeptides. Therefore, the observation of parallel changes in the Amide I and Amide II region favors the assignment of these changes to amide containing compounds. However, the Amide II band of Figure 2D is also unusually small compared to the Amide I in the same figure, which does not rule out a contribution from molecules other than polypeptides.

Alkyl ammonium ions can give absorption bands in the same region as the Amide I and Amide II bands. We can rule out a contribution from residual perovskite crystals because all their absorptions were below 1600 cm$^{-1}$ [18]. However, we cannot exclude that the main difference bands in Figure 2C arise from the methylammonium ions themselves, which may have been adsorbed by the cell, although they should have been removed by the extensive washing steps following fixation.

The diagram in Figure 2D shows the increase of the 1627 cm$^{-1}$ band over time relative to the 1652 cm$^{-1}$ band, measured on samples collected at different days. The band starts appearing within two days after exposure and rapidly increases over time. The slow time course of the increase disfavors the accumulation of methylammonium iodide as the origin of the 1627 cm$^{-1}$ absorption and suggests a mechanism that relies on biochemical changes in response to exposure. One important factor is the role of fixation chemistry on the final composition of the sample. While fixation preserves the morphology and part of the subcellular architecture of the cells, it also alters their chemical composition and the conformation of biological macromolecules. Molecular products of fixation should also be considered for the interpretation of spectroscopic data.

To clarify the assignments and exclude at least some possibilities, we analyzed the spectra using 2D correlation spectroscopy (2DCOS). The analysis allows us to identify groups of bands that can arise from the same compound or to exclude the ones that cannot by looking at their correlation properties. In standard applications, 2DCOS is applied to the spectra of samples that evolve as a function of time or as a function of some physical variable such as temperature or pressure [19]. We have previously applied this method to the assignment of bands in complex cellular spectra evolving in time [15,20]. In the present application, we used a recently introduced variation of the technique by reporting correlations as a function of position in the space of the measured spectrum [21]. We used such correlations to identify bands that arise from the same species as a way to confirm or disprove their assignment to specific molecules. For such purposes, we applied the 2DCOS algorithm to the set of spectra collected by mapping a single cell. The resulting synchronous and asynchronous spectra for a cell exposed to MAPbI$_3$ are shown in Figure 3. Bands that arise from the same molecular species or from species with the same spatial distribution must show synchronous changes (i.e., display cross correlation peaks (outside of the diagonal) with the same sign in the Synchronous plot). They must also display weak or absent correlation peaks in the Asynchronous plot, although the latter are often difficult to confirm because of band overlap. This is true for bands that arise from the same compound, but also for bands from different compounds that have the same spatial distribution. The synchronous plot in Figure 3A shows the presence of a peak multiplet at 1655, 1627, 1548, and 1515 cm$^{-1}$, marked with crosses along the diagonal, together with the corresponding correlation peaks outside of the diagonal. The positive value of the correlation peaks (red color) indicates that the peaks display an in-phase correlation. The same peak multiplet shows a lack of or minor correlation in the asynchronous plot of Figure 3B, indicating that these peaks arise from the same compound or from multiple compounds with the same spatial distribution. The correlation is consistent with, but not exclusive to, the assignment of bands in this multiplet to the Amide I and Amide II pairs [19].

In contrast, the opposite correlation was observed in the Synchronous plot of Figure 3A between the 1627 cm$^{-1}$ band and the carbonyl band at 1715 cm$^{-1}$, indicating that they arise from different molecular species. The observation excluded an assignment to molecules such as glycosaminoglycans or other polysaccharides that contain both amide and ester functions (although an assignment to their hydrolysis products is possible, as discussed later). It would be useful to assess the correlation between the 1627 cm$^{-1}$ band and the 2855 cm$^{-1}$/2920 cm$^{-1}$ bands in the CH$_2$ stretching region (not shown) that are typically assigned to long alkyl chains in lipids. However, the large difference in wavelength between the two regions means that the diffraction-limited focal spots differ in size by a factor of two and effectively probe different regions of the sample, thus altering the significance of the correlations.

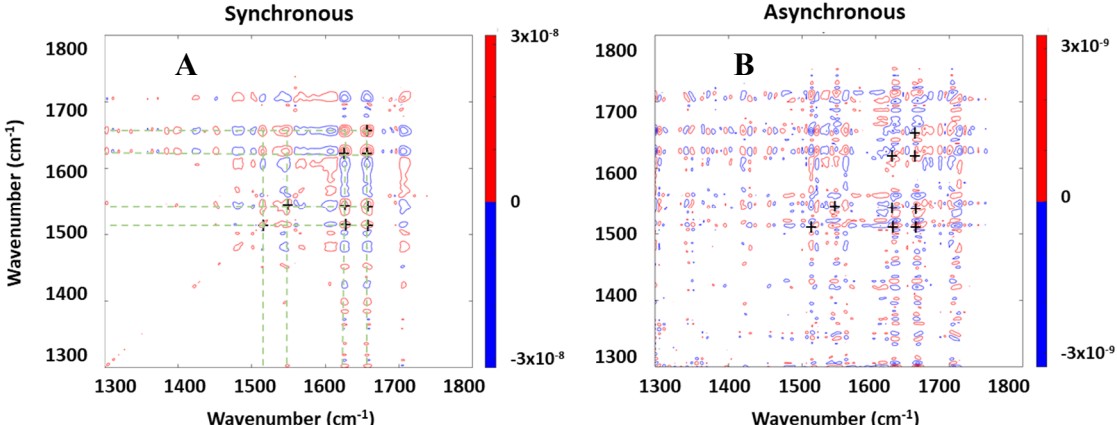

**Figure 3.** 2D correlation spectroscopy analysis of the spectral changes in a single SH-SY5Y cell exposed to 100 μg/mL of MAPbI$_3$ for five days. (**A**) Synchronous plot. The crosses identify the main peaks under discussion, at 1655, 1627, 1548, and 1515 cm$^{-1}$, along the diagonal, and the corresponding cross-peaks. The dashed green lines are used to guide the eye to the wavenumber scale. The Z scale (correlation) is in arbitrary units. Positive values of the correlation peaks (red color) indicate in-phase correlation; negative values (blue color) indicate opposite-phase correlation. (**B**) Asynchronous plot. The crosses identify the same locations marked in the synchronous plot.

An Amide I band in the proximity of 1620–1630 cm$^{-1}$ is often assigned to proteins in the amyloid fold, with a β-sheet structure supported by extended intermolecular hydrogen bonding. This fold is characterized by a two-component Amide I band, with a stronger component around 1620–1630 cm$^{-1}$ and a weaker one around 1660–1690 cm$^{-1}$ [22]. In the 2DCOS Synchronous plot of Figure 3A, we could not see correlation peaks involving a band in this position, suggesting that this is not a protein in the amyloid fold. A weak band is seen at 1678 cm$^{-1}$ in the difference spectra of Figure 2, but the lack of a synchronous correlation and appreciable asynchronous correlation suggests that the band is unrelated to the species that absorb at 1627 cm$^{-1}$ and 1655 cm$^{-1}$.

The correlation between bands at 1627, 1515, and 1548 cm$^{-1}$ suggests that the band may arise from a polypeptide. However, the intensity of the band at 1627 cm$^{-1}$, relative to the one at 1655 cm$^{-1}$, is surprising. After five days of exposure, the two bands became comparable, indicating that the number of amide bonds absorbing at 1627 cm$^{-1}$ is of the same order of magnitude as that of all other cellular polypeptides. This would represent a large change in the conformation of cellular proteins associated to the cell or the extensive expression of new proteins. It would be surprising for any known individual intracellular protein to account for such intensity.

An alternative assignment that is consistent with the 2DCOS spectra of Figure 3 is that the band at 1627 cm$^{-1}$ may arise from the hydrolysis of amide containing molecules. Such a reaction would lead to the formation of amine groups, with absorption in the proximity of 1620 cm$^{-1}$, and of a carboxylic acid group, which would absorb approximately in the 1550 cm$^{-1}$ region when deprotonated and in the 1700 cm$^{-1}$ region when protonated.

IR mapping of the 1627 cm$^{-1}$ band throughout an SH-SY5Y cell is shown in Figure 4. Mapping confirms that the band arises from molecules located within the cell itself or on the cell surface, and not from a deposit in its surroundings. The band is clearly absent in cells that were not exposed to metal salts. When present, it appears to track the distribution of the 1652 cm$^{-1}$ band. Unfortunately, the diffraction-limited resolution of the IR maps, approximately 7 µm at these wavenumbers, does not allow us to associate the band to any specific subcellular structure. Nonetheless, IR maps exclude the possibility that the band arises from an extracellular deposit.

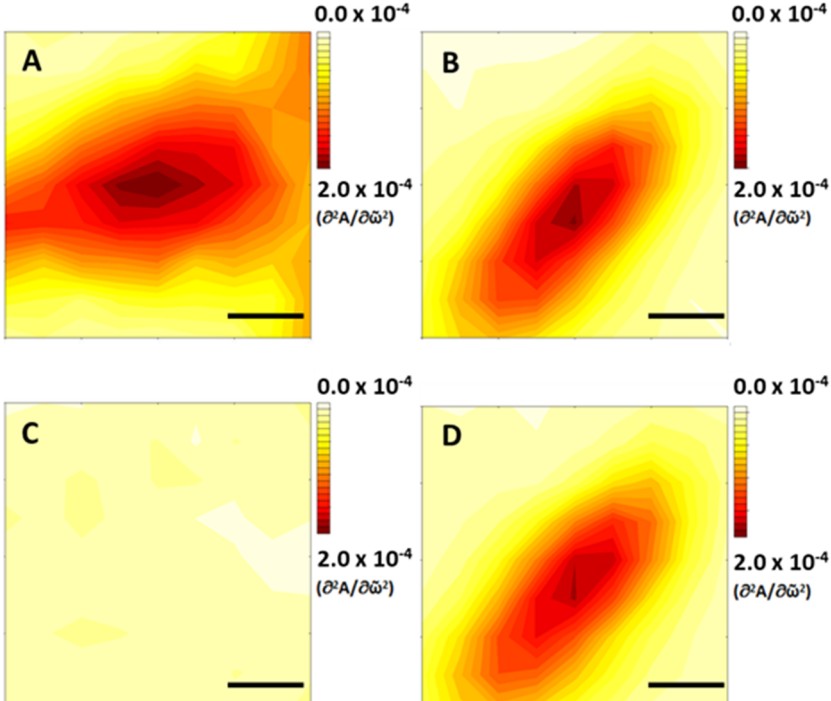

**Figure 4.** Transmission IR absorption maps of SH-SYS5 cells. (**A**) Map at 1652 cm$^{-1}$ of an untreated SH-SY5Y cell following fixation. (**B**) Map at 1652 cm$^{-1}$ of an SH-SY5Y cell following treatment with 100 µg/mL MAPbI$_3$ and fixation. (**C**) Map at 1627 cm$^{-1}$ of the same SH-SY5Y cell as in A. (**D**) Map at 1627 cm$^{-1}$ of the same treated SH-SY5Y cell as in (**B**). All maps show the distribution of minus the intensity of the second derivative of absorbance at the given wavenumber value. The cells treated with MAPbI$_3$ show an intense absorption band at 1627 cm$^{-1}$ with approximately the same distribution as the 1652 cm$^{-1}$ component of the Amide I band. The 1627 cm$^{-1}$ band is absent in the untreated cell. Scale bar: 20 µm. Figure reproduced and modified from [6] (I.B. doctoral thesis).

Identifying proteins that could give rise to the observed conformational changes is of interest in understanding the biochemical origin of Pb$^{2+}$ and Sn$^{2+}$ toxicity. Pb$^{2+}$ is known to bind tightly to Ca$^{2+}$ and Zn$^{2+}$ binding proteins under physiologically relevant conditions, thereby disrupting their function [23]. Several Zn$^{2+}$ and Ca$^{2+}$ proteins such as zinc finger proteins and calmodulin are regulatory proteins [24]. However, due to their relatively low abundance, regulatory proteins cannot account for the intensity of the observed spectroscopic changes.

Shelton et al. [25,26] and Gonick [27,28] reported the induced expression of a protein that associates with lead in forming intranuclear inclusion bodies in neuroblastoma cells exposed to lead. They also reported the constitutive expression in neural cells of a protein leading to inclusion body formation, following chronic lead exposure. The identity of the protein has not been definitively established and may also correspond to more than one polypeptide. It has been proposed that the protein or proteins accomplish a protective function by immobilizing lead in an insoluble form and prevent its interaction with other cellular components. Such a mechanism would be consistent with the de novo synthesis of large amounts of protein and explain the presence and intensity of the band at 1627 cm$^{-1}$. The

inclusion bodies formed by protein overexpression in engineered bacterial cells also display the IR spectrum of amyloids [29], in agreement with the hypothesis that inclusion bodies could be the origin of the spectral changes observed in our experiments. Despite this possibility, we failed to observe inclusion body formation in our samples using transmission electron microscopy (data not shown).

The appearance of a strong band around 1625 cm$^{-1}$ has been reported for eukaryotic cells undergoing apoptosis [30] and attributed to the induction of a β-sheet structure in cellular proteins. The band was formed without the exogenous addition of heavy metal ions. If the same molecular species were responsible for the appearance of this band in our samples, then the causal relationship with heavy metal ions may only be indirect. Heavy metal ions would be responsible for triggering apoptosis by inducing generally stressful conditions. The subsequent formation of the 1627 cm$^{-1}$ band would be associated with the apoptotic process itself via ex novo protein overexpression or via the refolding of existing proteins.

An interesting possibility is that proteins of the pentraxin structural family may be involved in the formation of the observed aggregated β-sheets. Some of them have been detected as a component of amyloid plaques [31] and have been reported to form intramolecular β-sheet structures absorbing at 1627 cm$^{-1}$. Pentraxin expression has not been studied in the cell lines used in our study, however, their known biological and biophysical properties make them candidates as components of the intermolecular β-sheet aggregates reported in our study.

One additional possibility is that the band may arise from albumin, which is known to refold into a β-sheet structure under the effect of temperature and metal ions [32]. Albumin is a component of the culture medium, making it one of the most abundant single proteins in a cell culture. It is possible that the refolded protein associates to the cell or the membrane via an unknown mechanism, giving rise to the observed changes. However, albumin is also present in the media used for untreated cells, although heavy metals in untreated samples are limited to the nutrients present in the medium and are present only in low concentration. A hitherto undefined mechanism that involves the heavy metals must mediate its association to or its uptake by the cell to fully explain our results. The appeal of this hypothesis is that, because of its abundance in the medium used for sample preparation, albumin can fully explain the intensity of the 1627 cm$^{-1}$ absorption. 2DCOS analysis excludes the presence of amyloids, implying that a different secondary structure must be at the origin of this band if the aggregation of albumin is involved.

Poly-L-lysine displays a strong band at 1627 cm$^{-1}$ from the absorption of the amine group in the lysine residue. The band overlaps with the absorption from the Amide I band of the polypeptide backbone when the latter is folded into a β-sheet structure at neutral or alkaline pH [33]. The superposition of these two contributions creates the appearance that the Amide I to Amide II ratio is unusually high. Poly-L-lysine was used as an adhesion layer for cells in the preparation of our samples. It is conceivable that the chelation of metal ions may be involved in its aggregation and association to the cell surface or internalization. This explanation would rationalize both the intensity of the band at 1627 cm$^{-1}$ relative to the overall cellular absorption and to the 1540 cm$^{-1}$ Amide II band, where a contribution from lysine absorption is absent.

In addition to the proteins just discussed above, additional species that satisfy the constraints resulting from the 2DCOS analysis include abundant amine and amide-containing molecules that can survive the washing steps involved in cellular preparation, mostly polysaccharides, but also any smaller molecules that may be tightly associated with cellular macromolecules or be involved in crosslinking during fixation. One final possibility is that part of the multiplets in the 2DCOS spectra results from the amines (approx.1600–1630 cm$^{-1}$) and carboxylates (approx.1500–1580 cm$^{-1}$) produced by the hydrolysis of amide containing molecules including polypeptides, sphingolipids, or polysaccharides. Systematic biochemical tests and analysis are required to restrict the pool of candidates and reach a final identification.

## 3. Materials and Methods

### 3.1. Culture of Mammalian Cell Lines

A549 Type II human alveolar basal adenocarcinoma epithelial cells (ECACC catalogue no. 86012804) were cultured in Dulbecco's Modified Eagle's Medium (DMEM) medium (Life Technologies, Carlsbad, CA, USA) supplemented with 10% fetal bovine serum (FBS) and 1% Penicillin/Streptomycin (*P/S*). Human neuroblastoma SH-SY5Y cells (ECACC catalogue no. 94030304) were cultured in a DMEM/F-12 medium (Life Technologies) supplemented with 10% FBS and 1% *P/S*. Both cell lines were cultured at 37 °C in a humidified, 5% $CO_2$ atmosphere.

Cells were plated on calcium fluoride ($CaF_2$) optical windows transparent for IR light (Crystran, Poole, UK) at a density of 20,000 cells using poly-L-lysine as an adhesion layer. Before administration to cells, $MAPbI_3$ and $MASnI_3$ powders were suspended in culture medium (concentration of 100 μg/mL). Solutions were vortexed, left for five days at Room Temperature (RT), filtered with a 0.22 μm pore size syringe filter (Merck Millipore, Burlington, MA, USA) and stored at 4 °C until used. Cells were incubated for five days with the halide perovskite medium solutions, rinsed with warm PBS (1×), and fixed using increasing concentrations of ethanol (20%, 40%, 60%, and 80%). Fixed cells were rinsed three times with MilliQ $H_2O$, left to dry overnight in a laminar flow hood, then stored in a desiccator for a maximum of five days before the IR measurements.

### 3.2. Toxicity Studies

In vitro toxicity studies were performed by exposing plated cells to filtered solutions of $MAPbI_3$ (100 μg/mL) or $MASnI_3$ (100 μg/mL), prepared as described in the previous section. Toxicity was assessed by counting viable cells in a Neubauer chamber at given time intervals. Only attached cells displaying a clear cytoplasm were counted as viable.

### 3.3. IR Microscopy Measurements

We performed Fourier transform infrared microspectroscopy (FTIRMS) measurements on beamline U2B at the National Synchrotron Light Source (NSLS, Brookhaven National Lab, Upton, NY, USA), and on beamline SMIS at the SOLEIL synchrotron (Paris, France). We used a Thermo Nicolet Continuum IR microscope set-up coupled to a Thermo Nicolet Magna 860 FTIR (Thermo Nicolet Instrument, Madison, WI, USA). We used a clean area of the $CaF_2$ optical window for background collection, then selected cells for IR measurements using the visible optical path of the microscope. Spectra were collected in transmission mode by closing the confocal apertures to obtain a diffraction-limited spot in the mid-IR spectral region, and raster scanning individual cells. Spectra were then used individually or assembled into 2D maps representing IR absorption throughout the cell. For detailed analysis, spectra were converted to the 2nd derivative of absorbance. The averaged spectrum of a single cell was obtained by discarding spectra for which the 2nd derivative of absorbance fell outside of the range |0.00002|−|0.0002| and averaging the remaining spectra. From one to three cells were mapped for each condition under study. To obtain the plots in Figure 4D, the spectra of 100–150 single cells were averaged for each time point. The ratio of the 2nd derivative absorption at 1652 $cm^{-1}$ and 1627 $cm^{-1}$ ± σ (variance) is shown in Figure 4D. IR data processing and analysis were performed using the software packages Omnic version 7.2 (Thermo Fischer, Waltham, MA, USA) and OPUS version 7.5 (Bruker Optics, Ettlingen, Germany). Spectral diagrams were created using OriginPro 2018 (Origin Lab Corporation, Northampton, MA, USA). 2DCOS analysis was performed using the software package MATLAB (MathWorks, Natick, MA, USA), as described previously. [20]

## 4. Conclusions

We used FTIRMS to study the effect of exposure to $MAPbI_3$ and $MASnI_3$ on SH-SYS5 and A549 cells at the molecular level. Our observations show that the cellular response to $Sn^{2+}$ ions and $Pb^{2+}$ ions leads to similar changes in the cellular IR absorption spectra. Exposure to both compounds leads

to the appearance of a strong band at 1627 cm$^{-1}$ accompanied by weaker absorption bands. Our results suggest that the band at 1627 cm$^{-1}$ is associated with a general response to metal cation exposure, since it is displayed by unrelated cell lines and different metal ions. The present experiments do not allow us to clarify whether this is related to a specific protective mechanism or if it arises from a non-specific response, with only an indirect causal link to the presence of heavy metals. Cellular fixation chemistry may also be contributing to the observed spectroscopic response. It is also unclear whether the changes thus detected have a direct or inverse causal relationship to the underlying molecular events that lead to cellular toxicity.

As a novel methodological tool, we used 2DCOS analysis on spatially resolved spectra to corroborate or disprove possible assignments of this band. The analysis allowed us to rule out the accumulation of polypeptides in the amyloid β-sheet fold as a possible assignment, despite the similarity of peak position with the one characteristic for the Amide I band of these structures. The same analysis also ruled out that the band may arise from acyl lipids or from polysaccharides containing both amide and ester groups.

Remaining hypotheses include the cellular accumulation of amine or amide containing molecules including some polysaccharides and smaller molecules resistant to fixation, and the metal-driven refolding and uptake of poly-L-lysine and/or albumin used in sample preparation. The aggregation and refolding of cellular proteins is not completely ruled out, although it looks unlikely in light of the extent of the changes and the need to involve a major fraction of cellular proteins. Future research involving a much larger dataset and biochemical tests will be directed to confirm or disprove these hypotheses.

While the identity of the major product of exposure, absorbing at 1627 cm$^{-1}$, has not yet been defined, excluding its assignment to an amyloid deposit is an important conclusion. The current literature on the IR microscopy of cells and tissue conventionally uses an absorption band in this position to identify the presence of amyloid deposits. We now show that such an assignment is unreliable when based only on such band and in the absence of additional biochemical characterization. Multiple alternative assignments are possible, corresponding to very different biochemical interpretations.

From the methodological point of view, these results are of interest beyond the specific biochemical issue. They provide preliminary corroboration of the usefulness of 2DCOS in simplifying the interpretation of complex biological spectra when applied to stationary samples such as fixed cells and tissue. This application of 2DCOS has to date been restricted to samples undergoing dynamic changes such as living cells. However, fixed or dried biological systems comprise most of the samples that are currently analyzed by FTIRMS, thus greatly expanding the scope of the method.

**Author Contributions:** Conceptualization, L.Q. and L.F.; Methodology, L.Q., B.V. and L.F.; Formal analysis, L.Q.; Investigation, I.B., B.V., E.H. and L.F.; Resources, L.F.; Data curation, L.Q., I.B. and B.V.; Writing—Original draft preparation, L.Q. and B.V.; Writing—Review and editing, L.Q. and B.V.; Visualization, L.Q. and B.V.; Supervision, L.F.; Project administration, L.F.; Funding acquisition, L.F. All authors have read and agreed to the published version of the manuscript.

**Funding:** This project received funding from the Swiss National Science Foundation (grant 200020_156981) and ERC Advanced Grant PICOPROP (to L.F.). L.Q. was supported by the European Union's Horizon 2020 research and innovation program under the Marie Skłodowska-Curie (grant agreement no. 665778), managed by the National Science Center Poland as Polonez 2 (grant UMO-2016/21/P/ST4/01321), and by the National Science Center Poland OPUS 16 (grant UMO-2018/31/B/NZ1/01345).

**Acknowledgments:** The authors are grateful to Lisa Miller, NSLS, and Christophe Sandt, Soleil, for help with the IR microscopy measurements, and to Hilal Lashuel, EPFL, for laboratory access and experimental support. L.Q. is grateful to Theodora Zlateva, Syneos Health, for fruitful discussion and comments.

**Conflicts of Interest:** The authors declare no conflicts of interest. The funders had no role in the design of the study; in the collection, analyses, or interpretation of data; in the writing of the manuscript, or in the decision to publish the results.

## Appendix A

To assess overall cellular toxicity of MAPbI$_3$ or MASnI$_3$, we exposed human SH-SY5Y and A549 cell lines to the filtered solutions of the perovskites and we counted the surviving cells at successive

time intervals. The results are summarized in Figure A1. These show a generally decreasing survival rate for both cell lines when exposed to either perovskite. Toxicity effects appear more marked for the SH-SY5Y cell line, which appear to shrink and detach following exposure. In contrast, A549 cells, while showing reduced mortality, also display morphological changes such as enlarged dimensions and multiple nuclei, and alterations of the cell cycle following exposure to both $MAPbI_3$ and $MASnI_3$. The different response of the two cell lines has already been reported when investigating the toxicity of $MAPbI_3$ [2] and is now confirmed for $MASnI_3$.

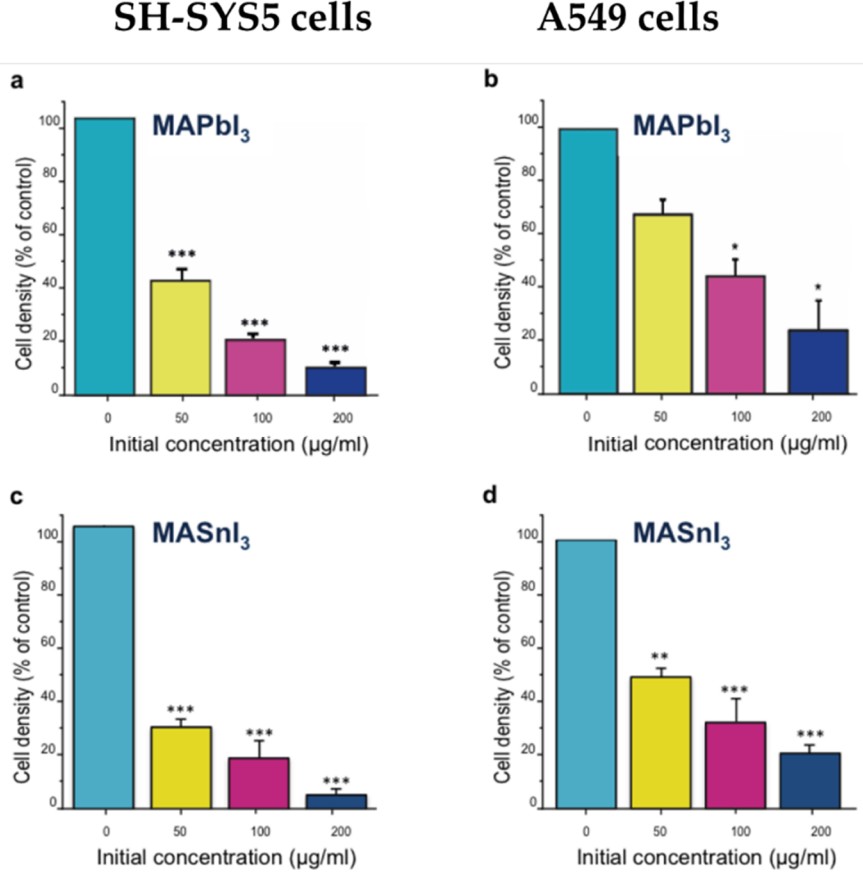

**Figure A1.** Quantification of living cells upon exposure to $MAPbI_3$ and $MASnI_3$. (**a**,**c**) SH-SY5Y neuroblastoma cells; (**b**,**d**) A549 lung epithelial cells. Histograms show the average of triplicate measurements. Bars are means ± σ. The histograms show an average of at least three independent repeats. Bars are means ± σ. One-way ANOVA tests followed by Tukey-Kramer post-hoc tests were performed (non-treated vs. MAPbI3 or vs. MASnI3 treated conditions), * $p < 0.01$, ** $p < 0.005$, *** $p < 0.0005$. Panel b is reproduced from [2] with permission from the Royal Society of Chemistry. Panels (**a**–**d**) are reproduced from [6] (I.B. doctoral thesis).

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
