# Peer review of "Infrared and 2-Dimensional Correlation Spectroscopy Study of the Effect of CH3NH3PbI3 and CH3NH3SnI3 Photovoltaic Perovskites on Eukaryotic Cells"

_molecules, doi:10.3390/molecules25020336_

Round 1

Reviewer 1 Report

I have the following major comments:

No doubt that the manuscript deserves publishing since it presents a new finding. However, there is still lack of conviction that this finding is correctly proven. The spectroscopic evidence and respective conclusions need, at least, additional testing e.g.by ecotoxicity biotests (there are plenty of them) in order to try to connect the spectroscopic results with some biological effects. Maybe the manuscript should be offered for present as a short communication instead as full paper. 

Author Response

Response to Reviewer 1.

I have the following major comments:

No doubt that the manuscript deserves publishing since it presents a new finding. However, there is still lack of conviction that this finding is correctly proven. The spectroscopic evidence and respective conclusions need, at least, additional testing e.g.by ecotoxicity biotests (there are plenty of them) in order to try to connect the spectroscopic results with some biological effects. Maybe the manuscript should be offered for present as a short communication instead as full paper. 

We have added the results of cytotoxicity tests performed by cell counting, which were already available. The toxicity experiments are now described in new Appendix A, also including one new figure. They are also mentioned in the text (approx. lines 85-87).

We leave it to the Editor to decide whether it is more appropriate to classify the work as a full article or as a communication. Either option is acceptable to us.

Reviewer 2 Report

Authors present Fourier Transform InfraRed Spectro-Microscopy data of biological cells under the effect of perovskites containing lead or tin. Effects of MAPbI3 (methylammonium lead iodide) and MASnI3 (methylammonium tin iodide) were investigated using spatially resolved 2-dimensional Correlation Spectroscopy (2DCOS). The work was interesting and the study offers valuable insights into molecular level changes with the inorganic interactions of X-perovskites used. I enjoyed the discussion on pages 7-8 for the possible fates of Pb2+ and Sn2+ and associated biological reasoning. However, I have some questions on the presented figures (listed below).

How were the control experiments done for the fixation of the cells in FTIR-MS experiments (or from previous literature)? Fixation processes often induce complex biological intermediates that can have specific interaction with the MAPbI3/MASnI3. Hence, I think the discussion of the results should mention the challenges of FTIR-MS in fixed cells. In all fairness, this is done in the conclusion section, but discussion of this problem would allow readers to keep an open mind on the biochemical scenario. I believe it would be easier on the reader if authors could mention in the results (Line 80) that the methods are described later (Line 323). What is the spatial resolution of the mentioned system? (line 236) Figure 2: How was the error bar on panel D obtained? Is it calculated among pixels or samples?
Figure 3: color/scale bar is unreadable. Would it be possible to tag the crosses on to the axes in the synch-mode? With high binning used in the 2DCOS graphs, it is hard to identify the position of the crosses shown.
Figure 4: some panels have an axis, some don't. Typos:
Line 15 : "organic inorganic" -> "organic/inorganic"
Line 65 : "the recent" -> "recent"
Line 68 : "one these" -> "one of these"
Line 88/201/336: "diffraction limited" -> "diffraction-limited"
Line 225 : "a new" -> "new"
Line 286: "present also" -> "also present"
Line 342: "was performed" -> "were performed"

Apart from these comments, I find the work interesting for the implementing new methods of analysis to interpret cellular changes in a multi/bulk-molecular level using FTIR microscopy. I believe this work present an enhanced characterization method for perovskites interaction in a biological sample.

Round 2

Reviewer 1 Report

The revised version of the manuscript fulfills the requirements and the suggestions of the reviewer. The addition of cytotoxicity test values improves substantially the study and could be used as confirmation of the idea of the authors.

I recommend acceptance of the manuscript in its revised version as a short communication.

Reviewer 2 Report

The authors have corrected the MS and improved the content and quality of the presentation. The work is interesting to the FTIR imaging community and I find a promising future for this work. I appreciate the authors' continuous effort in this field and I hope to see this work published.